# Simultaneous Quantification and Pharmacokinetic Study of Nine Bioactive Components of *Orthosiphon stamineus* Benth. Extract in Rat Plasma by UHPLC-MS/MS

**DOI:** 10.3390/molecules24173057

**Published:** 2019-08-22

**Authors:** Zili Guo, Bo Li, Jinping Gu, Peixi Zhu, Feng Su, Renren Bai, Xianrui Liang, Yuanyuan Xie

**Affiliations:** 1Key Laboratory for Green Pharmaceutical Technologies and Related Equipment of Ministry of Education, Zhejiang University of Technology, 18 Chaowang Road, Hangzhou 310000, China; 2College of Pharmaceutical Science, Zhejiang University of Technology, 18 Chaowang Road, Hangzhou 310000, China

**Keywords:** *Orthosiphon stamineus* Benth., bioactive constituents, UHPLC-MS/MS, rat plasma, pharmacokinetic study

## Abstract

*Orthosiphon stamineus* Benth. (OS) is a traditional folk medicine for the treatment of kidney stones and other urinary tract diseases. In this study, a rapid and sensitive Ultra high-performance liquid chromatography (UHPLC)-MS/MS approach was established and validated for the simultaneous quantification of nine bioactive components in rat plasma. The nine components from OS extract detected in rat plasma were danshensu, protocatechuic acid, caffeic acid, rosmarinic acid, salvianolic acid A, salvianolic acid B, cichoric acid, sinensetin and eupatorin. After liquid-liquid extraction with ethyl acetate, the plasma samples were subjected to a triple quadrupole mass spectrometer employing electrospray ionization (ESI) technique and operating in multiple reaction monitoring (MRM) with both positive and negative ion modes. The standard curves showed good linear regression (*r* > 0.9915) over the concentration range for the nine analytes. The inter-day and intra-day precision and accuracy were found to be within 15% of the nominal concentration. The recovery and stability of nine compounds were all demonstrated to be within acceptable limits. The approach was successfully applied to investigate the pharmacokinetic analysis of the nine bioactive components after oral administration of OS extract in rats.

## 1. Introduction

Natural products have been an exemplary source of new drugs, and many of the currently available medicines have been directly or indirectly derived from them [1]. *Orthosiphon stamineus* Benth. (OS), belonging to Labiatae (family), is widely distributed in China and cultivated at different regions in Malaysia. The dried whole plant of OS, also named “Shen Cha”, is a traditional folk medicine for the treatment of kidney stone and other urinary tract diseases [2]. OS contains several bioactive constituents, such as terpenoids and phenolic compounds, but one of the most important classes of the bioactive constituents is the phenolic group [3]. The reported therapeutic effects of OS, such as anti-inflammatory [4], diuretic [5], hypouricemic [2], antidiabetic [6], anti-hypertensive [7], anti-oxidant [8], hepatoprotective [9] and antiproliferative [10] activities ascribed mainly to the phenolic compounds. For example, protocatechuic acid (PCA), danshensu (DSS), salvianolic acid A (Sal A) and salvianolic acid B (Sal B), the main water-soluble components in OS exert various therapeutic activities in improving the renal function of rats with renal failure [11], eliciting anti-proliferative and pro-apoptotic effects in cancer cells [12], a removal of free radicals [13], as well as treating Alzheimer’s disease [14]. Sal B has already been successfully used for the treatment of coronary vascular diseases in clinical case [15]. Rosmarinic acid (RA), and cichoric acid (CA), as caffeic acid (CAA) derivatives, have been reported that displayed several bioactive effects, such as anti-inflammatory [16], antioxidant [17], and preventing insulin resistance activities [18]. In addition to these phenolic acids, the methoxylated flavones including sinensetin (SIN) and eupatorin (EUP) are also detected in OS. Furthermore, they also display various bioactivities including diuretic activity in rats [5], cell growth inhibition [19] and apoptosis induction preferentially in cancer cells [20].

In view of these various and potent pharmacological activities of these bioactive components in OS, increasing interests are focused on the analyses and identification of polyphenols in vitro and in vivo to understand the possible pharmacological effects. Although several analytical methods based on HPLC-UV [21], high-performance thin-layer chromatography (HPTLC) [22] and UHPLC-MS/MS [23] have been established for quantitative analysis of the major active constituents or to evaluate the pharmacokinetic profiles of flavonoids in rat plasma, the pharmacokinetic behaviors of PCA, DSS, CAA, RA, CA, Sal A and Sal B of OS remain unknown.

Therefore, the major purpose of this study was to develop a rapid and sensitive UHPLC-MS/MS method for the simultaneous quantification of nine polyphenols in rat plasma. Meanwhile, the approach was fully validated and successfully applied to the pharmacokinetic study in rats of these nine polyphenols after oral administration of OS extract. The obtained results indicated that the established approach was efficient and useful for comprehensive pharmacokinetic analyses of these polyphenols.

## 2. Results and Discussion

### 2.1. Method Development

The mobile phase played a key role in the acquisition of good chromatographic behavior and appropriate ionization. Compared with methanol as the organic phase, acetonitrile resulted in lower background noise, lower column pressure, and better peak shape. To obtain good peak shape and high detection sensitivity, different amounts of formic acid and acetic acid were added to the aqueous phases. When 0.05% formic acid was added to the aqueous phase, better peak shape and higher detection sensitivity were found. Therefore, acetonitrile containing 0.05% formic acid was chosen as the mobile phase.

To improve sensitivity, mass detection of the analytes was evaluated by the multiple reaction monitoring (MRM) scanning type in both positive and negative ion modes. A greater sensitivity was achieved for SIN and EUP in the positive ion mode while stable and intense signals of PCA, DSS, CAA, RA, CA, Sal A, Sal B and Internal standard (IS) were observed in the negative mode. Some mass spectrometric parameters, including collision voltage (CV), collision energy (CE) and different multiple reaction monitoring (MRM) transitions for nine analytes and IS were evaluated and optimized (shown in Appendix A). As shown in Figure 1, the selected quantitative precursor and product ions pairs were *m*/*z* 153.122→391.0576, 197.1486→135.1198, 179.1380→107.0708, 359.1803→161.1187, 373.2166→343.2086, 345.2492→312.1695, 473.1117→311.1818, 493.1532→295.1926, 717.1853→519.2154 and 321.1081→152.0853 for PCA, DSS, CAA, RA, SIN, EUP, CA, Sal A, Sal B and IS, respectively.

In a preliminary study, considering that solid-phase extraction method was tedious, time-consuming and cost-expensive relatively, the plasma sample pretreatment methods including protein precipitation with methanol and acetonitrile and liquid-liquid extraction (LLE) with ethyl acetate was tried. Results showed that the LLE with ethyl acetate presented the maximum recovery for the nine compounds and minimal matrix effect. Thus, the LLE with ethyl acetate was selected as the pretreatment method of plasma samples.

### 2.2. Method Validation

#### 2.2.1. Specificity and Selectivity

Figure 1 displays the typical MRM chromatograms of blank plasma (A), plasma spiked with the nine analytes and IS (B), and the samples collected from rats after oral administration of the OS extract (C). The retention time of DSS, PCA, CAA, RA, CA, Sal B, Sal A, SIN, EUP and IS were 1.78, 2.47, 3.77, 4.85, 4.95, 5.06, 5.33, 6.11, 6.38, and 5.12 min, respectively. No obvious endogenous interferences were observed at the same retention time of the nine analytes in blank plasma chromatograms.

#### 2.2.2. Linearity, Precision and Accuracy

The linear ranges, regression equations, and correlation coefficients of the nine analytes are shown in Table 1. The calibration curves showed good linearity with correlation coefficients (*r*) higher than 0.9915. Y is peak area; X is ng/mL. The intra-day and inter-day precisions (relative standard deviation, RSD) for PCA, DSS, CAA, RA, SIN, EUP, CA, Sal A and Sal B at three different levels with acceptable precisions and accuracy (Table 1) according to the U.S. Food and Drug Administration (FDA) bioanalytical method validation guidance [24]. The accuracies (relative error, RE) in the samples ranged from −13.60% to 14.80% (Table 2). These values were within the acceptable range, and the method was reproducible and reliable.

#### 2.2.3. Extraction Recoveries and Matrix Effect

The extraction recoveries and matrix effects of PCA, DSS, CAA, RA, SIN, EUP, CA, Sal A and Sal B from rat plasma are shown in Table 3. The recovery rates of the investigated analytes at three concentration levels ranged from 46.98% to 95.11%, with SD value less than or equal to 13.79%. The matrix effects of the nine analytes of interest varied from 67.62% to 125.76%, with RSD values less than 12.77%, 11.17%, and 5.13% at low, medium and high concentration respectively.

#### 2.2.4. Stability

Nine analytes were found to be stable in rat plasma after short-term storage at 25 °C for 24 h (RE: −6.40–12.67%, RSD ≤ 14.53%), −80 °C for 20 days (RE: −8.23–13.47%, RSD ≤ 9.54%), and after three freeze-thaw cycles at −80 °C (RE: −15.63–13.00%, RSD ≤ 9.61%). The results showed that the plasma samples were stable during normal experimental conditions (Table 4).

### 2.3. Pharmacokinetic Study

The validated UHPLC-MS/MS approach was applied to investigate the pharmacokinetic profiles for simultaneous determination of the nine analytes (PCA, DSS, CAA, RA, SIN, EUP, CA, Sal A and Sal B) in rats after oral administration of the OS extract at a dose of 10 g/kg. The mean plasma concentration-time profiles of PCA, DSS, CAA, RA, SIN, EUP, CA, Sal A and Sal B are shown in Figure 2. In addition, the main plasma pharmacokinetic parameters including AUC_(0–t)_, AUC_(0–∞)_, *C*_max_, *T*_max_, t_1/2z_ and MRT_(0–t)_ are listed in Table 5.

Except for DSS, CA and Sal B, the values of *T*_max_ of PCA, CAA, RA, SIN, EUP and Sal A were within 0.71 h after oral administration. It showed that these six analytes were absorbed quickly. DSS, CA and Sal B had a shorter absorption. Four analytes, including PCA, CAA, SIN and EUP, showed a relatively short t_1/2z_ (<1.89 h), indicating their rapid elimination. SIN had a shorter t_1/2z_ (0.59 h) than PCA (1.59 h), CAA (1.89 h) and EUP (1.13 h). DSS, RA and CA had a higher AUC_(0-t)_, indicating their better absorption. Meanwhile, SIN and EUP had a lower AUC_(0–t)_, maybe due to the low contents in the OS extract.

## 3. Materials and Methods

### 3.1. Chemicals and Reagents

OS was collected from Yunnan Kunming in June 2017 and stored at room temperature until analysis. The reference standards of PCA, CAA, RA and chloramphenicol (CHL, IS) were purchased from Aladdin Chemistry (Shanghai, China). DSS was purchased from National Institutes for Food and Drug Control (Beijing, China). CA, Sal A, Sal B, SIN and EUP were purchased from Chengdu Weikeqi Bio-Technology Co., LTD (Chengdu, China). L-Ascorbic acid (VC) was purchased from Shanghai Macklin Biochemical Co., Ltd. (Shanghai, China). The purities of all the reference standards were higher than 97%. HPLC grade acetonitrile and methanol were supplied by Merck (Darmstadt, Germany). Formic acid of HPLC grade was supplied by Aladdin Chemistry (Shanghai, China). Ultrapure water was produced by Barnstead TII super Pure Water System (Thermoscientific, Massachusetts, USA). All other analytical grade chemicals used in this experiment were purchased from Yongda Chemical Reagent Company (Tianjin, China). The chemical structures of these compounds are shown in Scheme 1.

### 3.2. Preparation of OS Extract

The dried herb of OS was powdered to a homogeneous size in a mill, passed through a 50-mesh sieve. A total of 250 g powder was refluxed twice with water (1:15, *w*/*v*) for 2.5 h and (1:10, *w*/*v*) for another 2.5 h. The mixed extract was evaporated under vacuum at 60 °C to remove water. Then it was dissolved in ethanol to remove polysaccharides and starch. After standing at room temperature for 24 h, the precipitate was removed by filtration. Then the filtrate was evaporated under vacuum at 45 °C to dryness to yield the OS extract (23.4 g). The contents of nine components in OS extract were measured quantitatively by an external standard assay using the same chromatography conditions as described below (in Section 3.3). The OS extract contained 0.34 mg/g PCA, 9.05 mg/g DSS, 3.73 mg/g CAA, 8.94 mg/g RA, 0.19 mg/g SIN, 0.26 mg/g EUP, 0.88 mg/g CA, 1.60 mg/g Sal A and 5.60 mg/g Sal B.

### 3.3. Equipment and UHPLC-MS/MS Conditions

Biological samples were performed with ACQUITY UPLC^®^ H Class system, which was coupled to an Xevo TQ-S micro triple quadrupole mass spectrometer (Waters, Milford, MA, USA) with an electrospray ionization (ESI) source and MassLynx^TM^ Workstation software (version 4.2, Waters, Milford, MA, USA).

Chromatographic separation was performed on a BEH Shield RP C18 column (100 × 2.1 mm, 1.7 µm, Waters, Milford, MA, USA) and the column temperature was maintained at 35 °C. The mobile phase, consisted of 0.05% formic acid aqueous solution (A) and acetonitrile (B), was delivered at a flow rate of 0.2 mL/min using the following gradient program: 0–2.5 min, 15–40% B; 2.5–4.0 min, 40–75% B; 4.0–6.0 min, 75–80% B; 6.0–6.1 min, 80–15% B; and then 6.1–10.0 min, 15% B for equilibration. The temperature of autosampler was set at 25 °C. The injection volume was 2 µL for samples.

Mass spectrometric detection was performed in both positive and negative ion modes. The optimized MS conditions were as follows: Capillary voltage 3.5 kV (positive) and 2.5 kV (negative), Source temperature 150 °C, Desolvation gas (N_2_) 800 L/Hr, Cone gas (N_2_) 50 L/Hr, Desolvation Temperature 450 °C.

### 3.4. Preparation of Calibration Standards and Quality Controls

The stock solutions of nine analytes were prepared in acetonitrile or methanol except for DSS, which was prepared in water, and stored at −80 °C. The IS stock solution (0.494 mg/mL) was prepared in methanol and diluted to a final concentration of 988 ng/mL with methanol before analysis. The mixed stock solution was prepared by serial dilution of each individual stock solution with methanol. Mix calibration standard samples containing PCA 8.70, 29.00, 145.00, 290.00, 580.00 and 2900.00 ng/mL, DSS 5.04, 16.80, 84.00, 168.00, 336.00 and 1680.00 ng/mL, CAA 6.48, 21.60, 108.00, 216.00, 432.00 and 2160.00 ng/mL, RA 3.06, 10.20, 51.000, 102.00, 204.00 and 1020.00 ng/mL, SIN 0.345, 1.150, 5.750, 11.50, 23.000 and 115.00 ng/mL, EUP 0.47, 1.567, 7.833, 15.67, 31.333 and 156.67 ng/mL, CA 4.95, 16.50, 82.50, 165.00, 330.00 and 1650.0 ng/mL, Sal A 3.69, 12.30, 61.50, 123.00, 246.00 and 1230.00 ng/mL, Sal B 4.20, 14.00, 70.00, 140.00, 280.00 and 1400.00 ng/mL were obtained by spiking the appropriate working solution into blank plasma.

In terms of the validation and pharmacokinetic study of the assay, three (low, medium and high) concentrations of the standard solution, including PCA (8.70, 290.00 and 2900.00 ng/mL), DSS (5.04, 168.00 and 1680.00 ng/mL), CAA (6.48, 216.00 and 2160.00 ng/mL), RA (3.06, 102.00 and 1020.00 ng/mL), SIN (0.345, 11.50 and 115.00 ng/mL), EUP (0.47, 15.67 and 156.67 ng/mL), CA (4.95, 165.00 and 1650.00 ng/mL), Sal A (3.69, 123.00 and 1230.00 ng/mL), Sal B (4.20, 140.00 and 1400.00 ng/mL) were applied to be the quality control (QC) samples. The standard solutions and QC samples were extracted on each analysis day with the same processes for plasma samples prepared as the description below.

### 3.5. Sample Preparation

Plasma samples were removed from −80 °C storage and thawed at room temperature before processing. To a 100 µL aliquot plasma sample 10 µL IS (988 ng/mL) and 20 µL of VC (1.0 mg/mL) were added. Then, the mixture was acidified with 10 µL of 1M HCl and vortex-mixed with 1mL of ethyl acetate for 3 min. After centrifugation at 13,000 rpm for 15 min. The upper organic layer was transferred to a clean Eppendorf tube and evaporated to dryness under nitrogen at 35 °C. The residue was reconstituted in 200 µL of MeOH-H_2_O (50:50, *v*/*v*). After being vortex-mixed 1 min followed by centrifugation at 13,000 rpm for 5 min, the supernatant was transferred into an auto-sample vial and an aliquot of 2 µL of the sample solution was injected into the UHPLC-MS/MS system for analysis.

### 3.6. Method Validation

The validation of the UHPLC-MS/MS method was evaluated based on selectivity, linearity, the lower limit of quantification (LLOQ), precision, accuracy, matrix effect, recovery, and the stability of nine compounds in rat plasma following the FDA Guidance for industry on bio-analytical method validation procedures [24].

#### 3.6.1. Selectivity

Selectivity was assessed by comparing the MRM chromatograms of six individual blank plasma samples, blank plasma spiked with standards and IS, and representative plasma samples after oral administration.

#### 3.6.2. Linearity and Lower Limits of Quantification

Linearity curves were obtained by assaying standard calibration samples at six concentration levels. The linearity of each calibration curve was determined by plotting the peak-area ratio (Y) of nine analytes to the IS versus the concentrations (X) of analytes with weighted least square linear regression. The LLOQ determined based on the analyte response should be at least 10 times that of the blank response.

#### 3.6.3. Precision and Accuracy

Intra-day precision and accuracy were assessed by analyzing six replicates of the QC samples at low, medium and high concentration levels (8.70, 290.00 and 2900.00 ng/mL for PCA, 5.04, 168.00 and 1680.00 ng/mL for DSS, 6.48, 216.00 and 2160.00 ng/mL for CAA, 3.06, 102.00 and 1020.00 ng/mL for RA, 0.345, 11.50 and 115.00 ng/mL for SIN, 0.47, 15.67 and 156.67 ng/mL for EUP, 4.95, 165.00 and 1650.00 ng/mL for CA, 3.69, 123.00 and 1230.00 ng/mL for Sal A, 4.20, 140.00 and 1400.00 ng/mL for Sal B). Inter-day assessments were similarly conducted on three consecutive days. The precision was expressed as the relative standard deviation (RSD, %) and accuracy was defined as a relative error (RE, %) from the theoretical concentrations.

#### 3.6.4. Extraction Recovery and Matrix Effect

Recoveries of the nine analytes from plasma were calculated by comparing the peak areas of pretreated QC samples with those of post-extracted blank plasma samples spiked with the analytes at the same concentration. Matrix effects were determined at three QC levels by comparing the peak areas obtained from blank blood extract spiked with the six analytes to those of pure standard solutions containing the same amount of the analytes.

#### 3.6.5. Stability

The stability of analytes in rat plasma was assessed at three QC levels under various conditions. The short-term stability was evaluated by analyzing samples kept at 25 °C for 24 h. The freeze-thaw stability was assessed over three freeze-thaw cycles (−80 °C to 25 °C). The long-term stability was tested by analyzing samples after stored at −80 °C for 20 days.

### 3.7. Pharmacokinetic Analysis

Six SPF-grade male Sprague-Dawley (SD) rats, weighed 250 ± 20 g, were provided by the Zhejiang Academy of Medical Sciences (Hangzhou, China). Rats were kept in an animal room with an ambient temperature of 22 ± 2 °C, the relative humidity of 55 ± 5% with 12 h light/dark cycles and were observed for one week in the Experimental Animal Center of the Zhejiang Province (Hangzhou, China) before starting the experiments. They were fed with freely available food and water, and fasted with free access to water for 12 h before drug administration. The experimental protocols involving animals were strictly followed the Guide for the Institutional Animal Care and Use Committee of Zhejiang University of Technology Laboratory Animal Center (20190301036).

The OS extract was dissolved in 0.5% carboxymethyl cellulose sodium (CMC-Na) aqueous solution to give an apparent concentration of 0.8 g/mL for oral administration. Then the OS extract was administered to rats orally with a single dose of 10.0 g/kg. For each rat, retro-orbital blood samples (~ 0.30 mL) were obtained into heparinized 1.5 mL polythene tubes before drug administration and at different time points of 0.083, 0.25, 0.50, 0.75, 1.0, 2.0, 4.0, 6.0, 8.0, 10, 12 and 24 h post-dosing. Subsequently, the blood samples were centrifuged at 4,000 rpm for 10 min to separate the plasma. Finally, the plasma was transferred to clean 1.5 mL Eppendorf tubes and stored at −80 °C until analysis.

### 3.8. Data Analysis

Pharmacokinetic parameters of the analytes were calculated using the pharmacokinetic software DAS 3.2 Version (Bontz Inc., Beijing, China). Pharmacokinetic parameters including maximum plasma time (*T*max) and concentration (*C*max), half-life (t1/2z), area under the plasma concentration versus time curve from zero to last sampling time (AUC_0–t_) and mean residence time (MRT_0–t_), were calculated by a non-compartmental approach from experimental data with no interpolation. All data were expressed as mean ± SD.

## 4. Conclusions

The current study established and validated a simple and sensitive UHPLC-MS/MS approach for the simultaneous determination of nine bioactive components (PCA, DSS, CAA, RA, SIN, EUP, CA, Sal A and Sal B) in rat plasma. The established approach showed good linearity with high selectivity, sensitivity, precision, accuracy, acceptable matrix effect and consistent recovery according to the guidelines. The approach was successfully applied to study the plasma pharmacokinetics following oral administration of OS extract. The sample pretreatment procedure is straightforward and the analysis running time is short. This approach could provide a scientific basis for the application of OS.

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
