# Peer review of "Simultaneous Quantification and Pharmacokinetic Study of Nine Bioactive Components of Orthosiphon stamineus Benth. Extract in Rat Plasma by UHPLC-MS/MS"

_molecules, 2019, doi:10.3390/molecules24173057_

Round 1

Reviewer 1 Report

This is a surprisingly well-written manuscript describing the development of an analytical LCMS method with pharmacokinetic analysis of nine bioactive components of Orthosiphon stamineus Benth.

line
 50  "Except" is unclear; do you mean, "in addition to"?

109  provide the citation, [20]

112  Table 1 could be strengthened by providing a more complete description of the variables: Y is peak area; x is ng/mL; 
it is not clear why the intercept is not zero. why is there a significant peak area when there is no analyte? 

144  "indicating ... body." is redundant and can be deleted

144  Figure 2 is unacceptable. The labeling needs to be enlarged so that readers can understand the graphs.  The graphs can additionally be enhanced by optimizing the time axis for the data.   

206  "were applied to prepare the quality control" is unclear

210  delete "executed"

258  The rats were fasted for 12 h prior to administration of the OS extract.  However, it is not clear when food was returned.  This is important because at least two of your analytes (sorry I can't read which ones) show secondary peaks suggesting enterohypatic cycling when food is returned.  

264  Perhaps my greatest concern regards the blood sampling used in this study.  The authors describe collecting 12 blood samples of 0.3mL; for a 250g rat, sampling 3.6mL of blood is excessive, approaching 25% of total blood volume. At a minimum, the authors need to discuss this.

Overall, a nice manuscript with the major issue of blood volume and some minor correctable issues.   

Reviewer 2 Report

The authors developed an analytical method using UPLC-MS/MS to determine bioactive compounds of Orthosiphon stamineusextract in rat plasma. Although the developed method was appropriately validated according to the FDA guideline, the manuscript should be corrected for publication by considering the following comments:

Major comments

In Table 3, it is unclear why there are three different values for recovery and matrix effect per each compound. Did the author test recovery and matrix effect at three different concentrations? Please clarify the table. In section 3.2, how did author measure the content of each compound in the dried extract? If the authors measured the contents in the extract using the other method that is published by Guo et al (2019, JBPA; Reference 23), the reference should be cited in the main text. The first paragraph in section 3.3, check the grammar. Moreover, author described 2 ul of “plasma sample” was injected into the UPLC-MS/MS in section 3.3. It should be changed to “sample”. For rat experiment, the administered volume to rats was 12. 5 ml/kg (10 g/kg / 0.8 g/ml), which is larger than the generally acceptable injection volume (< 10 ml/kg). In Supplementary Table 1, why does each compound have two different MS transitions? Please describe why. If it is not necessary, remove one of MS conditions for compounds and combined the table with the main text.

Minor comments

In Table 1, replace “concentration range” to “calibration range”. In Figures 1 and 2, the legend for each compound is unreadable. Please clarify each panel or make a bigger legend for each compound. In line 210, there is a grammar error (executed removed). In line 268, “plasma was centrifuged~” should be changed to “blood samples were centrifuged~”.

Reviewer 3 Report

Title: Simultaneous quantification and pharmacokinetic study of nine bioactive components of Orthosiphon stamineus Benth. extract in rat plasma by UHPLC-MS/MS

Authors developed and validated an UHPLC-MS/MS method to determine nine compounds in rat plasma after oral administration of Orthosiphon stamineus Benth. extract. This manuscript was well organized and presented. There are some of minor comments.

1, Page 2, line 74: -0.05% should be “containing 0.05%”.

2, the resolution of figure 1 and 2 should be improved.

3, In table 1, the column of LLOQ can be deleted since it was presented in the concentration range.

4, Some of mass spectrometric parameters were missing, such as voltage, etc.

5, Page 9 line 215: MeOH-H2O should be “MeOH-H2O”

Round 2

Reviewer 2 Report

The manuscript is improved and fits for publication in Molecules.